# The MATS satellite: Limb image data processing and calibration

Linda Megner<sup>1</sup>, Jörg Gumbel<sup>1</sup>, Ole Martin Christensen<sup>1</sup>, Björn Linder<sup>1</sup>, Donal P. Murtagh<sup>2</sup>, Nickolay Ivchenko<sup>3</sup>, Lukas Krasauskas<sup>1</sup>, Jonas Hedin<sup>1</sup>, Joachim Dillner<sup>1</sup>, Gabriel Giono<sup>3</sup>, Georgi Olentsenko<sup>3</sup>, Louis Kern<sup>1</sup>, and Jacek Stegman<sup>1</sup>

**Correspondence:** Linda Megner (linda@misu.su.se)

Abstract. MATS (Mesospheric Airglow/Aerosol Tomography and Spectroscopy) is a Swedish satellite mission designed to investigate atmospheric gravity waves. In order to observe wave patterns MATS observes structures in the O<sub>2</sub> atmospheric band airglow (light emitted by oxygen molecules in the Mesosphere and Lower Thermosphere), as well as structures in noctilucent clouds which form around the Mesopause. The main instrument is a telescope that continuously captures high-resolution images of the atmospheric limb. Using tomographic analysis of the acquired images, the MATS mission can reconstruct waves in three dimensions and provide a comprehensive global map of the properties of gravity waves. The data provided by the MATS satellite will thus be 3-dimensional fields of airglow and NLC properties in 200-km-wide (across track) strips along the orbit at 70 to 110 km altitude. Adding spectroscopic analysis, by separating light into six distinct wavelength channels, it also becomes possible to derive temperature and microphysical NLC properties. Based on those data fields, further analysis will yield gravity wave parameters, such as wavelengths, amplitudes, phase, and direction of the waves, on a global scale.

The MATS satellite, funded by the Swedish National Space Agency, was launched in November 2022 into a 580 km sunsynchronous orbit with a 17.25 local time of the ascending node (LTAN). This paper accompanies the public release of the level 1b (v. 1.0) data set from the MATS limb imager. The purpose of the paper is to provide background information in order to assist users to correctly and efficiently handle the data. As such, it details the image processing and how instrumental artifacts are handled. It also describes the calibration efforts that have been carried out on the basis of laboratory and in-flight observations, and it discusses uncertainties that affect the dataset.

#### 1 Introduction

In recent decades, it has become increasingly evident that the different layers of the atmosphere, spanning from the surface to the ionosphere, are closely interconnected. One crucial factor contributing to these connections is the existence of atmospheric gravity waves (GW), which are buoyancy waves that carry substantial amounts of energy and momentum. The influence of GW on atmospheric circulation, variability and structure has generated substantial research interest, as scientists recognise the significance of the upper regions of the atmosphere in the climate system (e.g. Fritts and Alexander, 2003; Geller et al., 2013;

<sup>&</sup>lt;sup>1</sup>Department of Meteorology, Stockholm University, Stockholm Sweden

<sup>&</sup>lt;sup>2</sup>Earth and Space Sciences, Chalmers University of Technology, Göteborg, Sweden

<sup>&</sup>lt;sup>3</sup>School of Electrical Engineering, Royal Institute of Technology (KTH), Stockholm, Sweden

Smith, 2012; Alexander et al., 2010). As a result, climate models are now expanding to higher altitudes, necessitating accurate descriptions of the effects of GW.

25

55

Despite dedicated research efforts, our understanding of global wave characteristics and momentum deposition remains incomplete. This knowledge gap primarily stems from observational constraints. Firstly, gravity waves are relatively smallscale phenomena (ranging from a few kilometres to several hundred kilometres); thus, high-resolution observations are needed to detect them. Secondly, quantification of the most important effects that gravity waves have on the general circulation of the atmosphere is only possible if gravity wave momentum and energy fluxes can be estimated. This either requires highresolution measurements of all three wind components (e.g. Fritts and Alexander, 2003), or observations of three-dimensional temperature structures that allow full characterisation of individual waves (e.g. Ern et al., 2015), or some combination of the two. Such measurements have been realised using ground-based radar (e.g. Stober et al., 2013), combinations of lidar and airglow imaging (e.g. Cao et al., 2016), and various in-situ techniques (e.g. Vincent and Alexander, 2000; Podglajen et al., 2016; Smith et al., 2016), but all of these observations, by their very nature, lack global coverage. Global 3D temperature data are available from some nadir-viewing satellite missions (e.g. AIRS, Hindley et al., 2020), but they lack vertical resolution and hence miss large parts of the gravity wave spectrum. Limb sounding satellites generally have better vertical resolution (e.g. SABER, Russell III et al., 1999), but their poor horizontal resolution (especially along the line of sight) does not allow for the full wave characterisation necessary for gravity wave energy and momentum flux estimations. In lack of these, any estimate becomes heavily reliant on statistical assumptions (e.g. Ern et al., 2011). To perform a comprehensive analysis of gravity waves, it is desirable to obtain three-dimensional temperature retrievals that include spatial structures with horizontal wavelengths shorter than 100 km and vertical wavelengths shorter than 10 km (Preusse et al., 2008). Some limbsounders such as Aura-MLS Livesey et al. (2006) and OMPS Zawada et al. (2018) have improved their horizontal resolution using along track tomographic retrievals reaching on the order of 200 km resolution. The MATS satellite has been designed specifically to address the lack of knowledge of smaller scale gravity waves. Its mission is to capture three-dimensional wave structures in the Mesosphere and Lower Thermosphere (MLT) by observing two phenomena: noctilucent clouds and atmospheric airglow in the O<sub>2</sub> A-band. These phenomena serve as visual indicators of gravity waves. The primary instrument, the limb imager, acquires atmospheric limb images in six different wavelength intervals: two in the ultraviolet range (270-300 nm) and four in the infrared range (760-780 nm). The two ultraviolet channels target structures in noctilucent clouds, and the infrared channels capture perturbations in the airglow. Using several infrared channels that cover the different spectral sections of the atmospheric A-band, temperature information can be obtained.

To achieve high-resolution imaging, the MATS satellite employs an off-axis three-mirror telescope with free-form mirrors, designed to achieve high resolution and minimise stray light. The limb of the atmosphere is imaged onto six CCD channels, and wavelength separation is achieved using a combination of dichroic beam splitters and narrowband filters. CCD sensors with flexible pixel binning and image processing capabilities are employed for image detection.

In addition to limb images, the MATS satellite incorporates a nadir camera that continuously captures images of a 200 km swath below the satellite. This provides detailed two-dimensional pictures of airglow structures below the satellite. Furthermore, the satellite includes two nadir-looking photometers that measure upwelling radiation from the Earth's surface and

the lower atmosphere. This article describes the data processing applied to the limb images. The data from the supporting instruments will be described in separate publications.

The MATS mission is based on the InnoSat spacecraft concept (Larsson et al., 2016), a compact (60×70×85 cm) and costeffective platform for scientific research missions in low-earth orbit. The mission is funded by the Swedish National Space
Agency (SNSA) and has been developed as a collaborative effort between OHB Sweden, ÅAC Microtec, the Department
of Meteorology (MISU) at Stockholm University, the Department of Earth and Space Sciences at Chalmers University of
Technology, the Space and Plasma Physics Group at the Royal Institute of Technology in Stockholm (KTH), and Omnisys
Instruments. For more detailed information on the mission and payload, see Gumbel et al. (2020), and for the optical design of
the instrument, see Hammar et al. (2019) and Park et al. (2020).

The primary scientific data obtained from the MATS limb imager consist of two-dimensional images of  $O_2$  A-band airglow and noctilucent clouds. By employing tomographic retrieval techniques combined with spectral information, it becomes possible to derive three-dimensional atmospheric properties such as airglow volume emission rate, atmospheric temperature, odd oxygen concentrations, and noctilucent cloud characteristics. In order to turn the raw limb images into scientific data products, several levels of processing are required, with accompanying date products planned according to the following:

- Level 1a (L1a): Geolocated image frames with meta-data such as time, instrumental settings, and temperatures of the instrument.
- Level 1b (L1b): Calibrated images with tangent point altitude information added to the meta-data.
- Level 1c (L1c): Calibrated images on a unified angular grid across all channels.
- Level 2 (L2): The retrieved atmospheric quantities. These will include 3D tomographic retrievals of temperature, volume emission rates of the oxygen A-band, and volume scattering coefficients for NLCs.

This paper accompanies the first release of the Level 1b data product for data collected between February and May 2023. During May 2023, the reaction wheels used for attitude control onboard the satellite started malfunctioning, and the scientific operations came to a stop. Currently a new steering algorithm based on magneto-torquers is being developed. At the time of writing it is not clear how accurate the pointing will be in the future and if a continuation of the dataset can be obtained.

In this paper, the steps leading to the L1b data product are presented. Specifically, the geolocation, calibration, and image processing of the data from the MATS limb imager are described in detail, to facilitate a better understanding and proper use of the resulting data products. We begin with an overview of the limb instrument (Section 2), followed by an explanation of the downlink procedures to transmit and unpack the satellite data (Section 3), which generate the inputs for L1a. Section 4 details the mapping of the field of view. The processes of image handling and calibration are covered in Section 5. Section 6 addresses effects that are not corrected for in the L1b data product but will be handled in subsequent data products. The discussion on error estimation and uncertainty of the data product is found in Section 7, with examples from the L1b data images presented in Section 5.9. Conclusion and summary are given in Section 8.

90

75

#### 2 The MATS limb instrument

The limb instrument is a reflective telescope with different spectral channels, according to the optical layout shown in Figure 1. Light enters the telescope via a baffle, which is coated with ultra-absorbent black paint (Vantablack) to reduce the amount of straylight coming from outside the field of view. The telescope is comprised of three off-axis aluminium mirrors crafted by Millpond ApS, with free-form surfaces created through diamond turning. The focal ratio of the telescope is 7.3 and the effective focal length of the telescope is 261 mm. The field of view is  $0.91 \times 5.67^{\circ}$ , which at the tangent point corresponds to about 40 km in the vertical and 260 km horizontally. The final images cover a slightly smaller area as they are generally cropped to allow misalignment between the channels, as explained in Section 3. The field of view was chosen to be as large as possible in order to observe larger-scale atmospheric waves, while still enabling the high resolution (down to 0.25 km vertically) needed to retrieve the smaller-scale structures. Despite the advanced optical telescope, there is a trade-off between total field-of-view and imaging resolution.

Following the telescope, dichroic beam splitters and thin-film interference filters are used to separate the incoming light into different spectral channels, see Figure 1 and Table 1. Each of the instrument's six channels uses a combination of broadband filter ("Filter B" in the figure) to remove out-of-band signals and a narrowband filter ("Filter N" in the figure) that ultimately defines the transmitted wavelengths. In order to keep the optical setup within the InnoSat satellite platform envelope, the optical path is reflected using two folding mirrors.

As can be seen in Fig. 1 the optical path of the channels IR1, IR3, UV1 and UV2 contains an odd number of reflections, whereas the channels IR2 and IR4 contain an even number of mirrors. The result is that the channels IR1, IR3, UV1 and UV2 are mirrored relative to IR2 and IR4. This is taken care of in the postprocessing, where a horisontal flip is applied to the images taken by IR1, IR3, UV1, and UV2.

At the end of each channel the images are recorded using passively cooled backlit CCD sensors (Teledyne E2V-CCD42-10) which are read out by a CCD readout box (CRB) that sends the data to the on-board computer. The CRB settings and data readout, as well as the CCD voltage, can be controlled by the power and regulation units.

As the CCD is read out, the pixel rows are progressively shifted downwards. In many CCD-based instruments, shutters are used to prevent CCD pixels from being exposed during the readout procedure. However, in order to avoid the risks of moving parts in the satellite, it was decided not to use shutters on MATS. As a consequence, the image rows continue to be exposed during the readout shifting, resulting in image smearing, which is corrected for as part of the data processing.

# 3 Preprocessing

Each CCD detector in the limb instrument comprises 2048 columns and 511 rows. In nominal science mode, with multiple channels that capture images every few seconds, the data volume exceeds the downlink capacity. Consequently, the images must be binned, cropped, and compressed on the satellite before being transmitted to the ground. Information on the particular binning, cropping and compression that have been used is saved as meta-data with each image.

**Figure 1.** Optical layout of the MATS limb instrument. The telescope mirrors are denoted as M1-M3. The subsequent spectral selection unit comprises beam splitters (BS), broad (B) and narrow (N) filters, folding mirrors (FM), and the six CCD sensors.

# 3.1 Image binning and cropping

130

The instruments have two methods to perform pixel binning; on-chip binning and binning in the subsequent Field-Programmable

Gate Array processor (FPGA). The main channels IR1, IR2, UV1, and UV2 use a column binning of 40 and a row binning of

2, resulting in a sampling of approximately 5.7 km × 290 m at the tangent point, while the background channels IR3 and IR4

use a column binning of 200 and a row binning of 6, resulting in a resolution of approximately 29 km × 860 m.

The imaged area on each CCD is larger than the required field of view of the instrument, and thus larger than the full baffle opening. This is to allow for slight misalignment of the CCDs and leads to baffle vignetting on each side of the full image. To save downlink data rate, images are generally cropped both vertically and horizontally so that areas affected by baffle vignetting are removed. However, some of the area at the bottom of the image is kept despite being strongly affected by vignetting, since it is needed to correct for smearing effects; see Section 5.5. Throughout the mission's commissioning phase, the specific cropping

| Denotation   | Wavelength Selection                          | Out-of-Band Blocking                          |  |  |  |
|--------------|-----------------------------------------------|-----------------------------------------------|--|--|--|
| Filter N UV1 | CWL 270.0±0.5 nm, FWHM 3.0±0.5 nm             | OD4 down to 200 nm and up to 1000 nm          |  |  |  |
| Filter N UV2 | CWL 304.5 $\pm$ 0.5 nm, FWHM 3.0 $\pm$ 0.5 nm | OD4 down to 200 nm and up to 1000 nm          |  |  |  |
| Filter N IR1 | CWL 762.4 $\pm$ 0.3 nm, FWHM 3.5 $\pm$ 0.5 nm | OD4 down to 350 nm and up to 1000 nm          |  |  |  |
| Filter N IR2 | CWL 763.0 $\pm$ 0.5 nm, FWHM 8.0 $\pm$ 0.5 nm | OD4 down to 350 nm and up to 1000 nm          |  |  |  |
| Filter N IR3 | CWL 754.0 $\pm$ 1.0 nm, FWHM 3.0 $\pm$ 1.0 nm | OD4 down to 350 nm and up to 1000 nm          |  |  |  |
| Filter N IR4 | CWL 772.0 $\pm$ 1.0 nm, FWHM 3.0 $\pm$ 1.0 nm | OD4 down to 350 nm and up to 1000 nm          |  |  |  |
| Filter B UV1 | 266-274 nm: transmission > 50%                | 350-1000 nm: blocking at least OD1 in average |  |  |  |
| Filter B UV2 | CWL 302-306 nm: transmission > 50%            | 350-1000 nm: blocking at least OD1 in average |  |  |  |
| Filter B IR  | CWL 750-780 nm: transmission > 90%            | 250-720 nm: transmission 

Figure 2. The image coordinate system, showing ascension and declination in degrees. The pink dots mark the center of the individual images.

X

### 5 Single image processing and calibration

This section describes the corrections and calibration procedures that make up the L1a to L1b processing. The L1a product is the result of the preprocessing described in Section 3, and comprises decompressed 16 bit images, where each pixel represents

- the count numbers collected by the analogue-to-digital converter (ADC), along with complementary meta-data. The counts are determined not only by the incident light, but also by a number of instrumental artefacts, which must be characterised and compensated for. These will be presented in the subections below. Before the main L1a processing, artefacts that affect only a limited number of pixels need to be corrected. This is handled by the pre-calibration pixel correction, which consists of the following steps:
  - 1. Single event correction, which corrects for high-energy particle effects on the CCD (see Section 5.1).
    - 2. Correction of hot pixels, which are pixels that exhibit abnormally high dark current. They change over time, and are monitored and corrected independently (see Section 5.2).

The reader is referred to the corresponding sections for details on how these steps are performed. We now denote the counts recorded for the pixel x, y in the CCD channel i, after the above corrections have been applied, as  $S^i(x,y)$ . The purpose of the L1a processing is to translate  $S^i(x,y)$  to the actual incident radiance  $L^i(x,y)$  in photons  $m^{-2}s^{-1}sr^{-1}nm^{-1}$ ,

 $S^{i}(x,y)$  can be expressed as:

$$S^{i}(x,y) = f_{\text{nonlin}}^{i} \left( f_{\text{smear}} \left( S_{\gamma}^{i}(x,y) + S_{\text{dark}}^{i}(x,y) \right) \right) + S_{\text{bias}}^{i}. \tag{4}$$

Here  $S_{\gamma}^{i}(x,y)$  is the signal (in counts) from the light hitting the CCD pixel, and  $S_{\rm dark}^{i}(x,y)$  is the signal (also in counts) from the dark current (counts due to thermally exited electrons in the absence of light, see section 5.6).  $f_{\rm smear}$  is a function that describes the smearing introduced by exposure during the readout, and  $f_{\rm nonlin}^{i}$  is a function that describes any non-linearity in the signal processing such as saturation or amplifier distortion. Finally,  $S_{\rm bias}^{i}$  is the electronics bias.

The quantity we seek, the radiance  $L^i(x,y)$ , is related to  $S^i_{\gamma}(x,y)$  by the equation:

$$S_{\gamma}^{i}(x,y) = L^{i}(x,y) \cdot G^{i}(x,y) \cdot t_{\text{int}}^{i}, \tag{5}$$

where  $t_{\text{int}}^i$  is the integration time of channel i.

 $G^i(x,y)$  [counts  $\cdot$  ph<sup>-1</sup>  $\cdot$  m<sup>2</sup>  $\cdot$  sr  $\cdot$  nm] is the calibration factor, which relates the incoming light in a channel to the number of counts it produces in the detector, taking into account both the throughput of the instrument optics and the quantum efficiency of the CCD, as these two effects are neither possible nor necessary to separate.

The factor  $G^i(x,y)$  can be expressed as

$$G^{i}(x,y) = \frac{\Omega \cdot F^{i}(x,y)}{a \cdot r^{i}}.$$
(6)

Here,  $\Omega$  is the solid angle of a CCD pixel as described in Section 5.8. The factor a [ph·cm<sup>-2</sup>·nm<sup>-1</sup>/counts] represents the absolute calibration factor that relates the number of incident photons to the number of counts generated by incident light. In principle, a could be independently measured for each channel. However, in the case of MATS,  $a = a_{IR2}, a_{UV2}$  is directly determined only for two reference channels (IR2 and UV2) and then related to the other channels by relative

calibration (Section 5.8). This approach minimises the uncertainties in the relative difference between channels, leading to a smaller uncertainty in the temperature retrieval.  $r^i$  are thus the relative calibration factors that describe the sensitivity of a particular channel relative to the sensitivity of the reference channels, so that by definition  $r^{IR2} = r^{UV2} = 1$ . The factors  $a \cdot r^i$  are determined as the average response in a reference area  $A_{ref}$  in the middle of each CCD detector where there is no baffle interference (see Section 5.8). Finally, the dimensionless flat field factors  $F^i(x,y)$ , describe the nonuniformity of the detector sensitivity, by relating the response of the individual pixels on a specific CCD to the reference area of that CCD (see Section 5.7). Combining Eq. (5) with Eq. (6),  $L^i(x,y)$  can be expressed as

$$L^{i}(x,y) = \frac{S_{\gamma}^{i}(x,y)}{G^{i}(x,y) \cdot t_{\text{int}}^{i}} = \frac{a \cdot r^{i} \cdot S_{\gamma}^{i}(x,y)}{\Omega \cdot F^{i}(x,y) \cdot t_{\text{int}}^{i}},\tag{7}$$

where  $S_{\gamma}^{i}(x,y)$  according to equation (4) is

$$S_{\gamma}^{i}(x,y) = f_{\text{desmear}}\left(f_{\text{lin}}^{i}\left(S^{i}(x,y) - S_{\text{bias}}^{i}\right)\right) - S_{\text{dark}}^{i}(x,y). \tag{8}$$

In this context,  $f_{\text{desmear}}$  ideally serves as the inverse of  $f_{\text{smear}}$ , and it can be approximated by assessing the additional exposure each row has received, as described in Section 5.3.  $f_{\text{lin}}$  is a function that corrects for the non-linearity of the signal strength, i.e. if perfect the inverse function of  $f_{\text{nonlin}}$  (see Section 5.4).  $S_{\text{bias}}^i$  is the electronic bias, which is estimated using non-exposed pixels as explained in Section 5.3. The factors  $a, r^i, F^i(x,y)$  and  $S_{\text{dark}}^i(x,y)$  are determined in calibration experiments as described later in this section.

Finally inserting (8) into (7) yields the formula that describes the L1a to Llb processing relating the counts on the CCD  $S(x,y)^i$  to the actual incident radiance  $L^i(X,Y)$  in photons  $m^{-2}s^{-1}sr^{-1}nm^{-1}$ :

$$L^{i}(x,y) = \frac{1}{G^{i}(x,y)} \frac{f_{\text{desmear}}\left(f_{\text{lin}}^{i}\left(S^{i}(x,y) - S_{\text{bias}}^{i}\right)\right) - S_{\text{dark}}^{i}(x,y)}{t_{\text{int}}^{i}}$$

$$(9)$$

or

$$L^{i}(x,y) = a \cdot r^{i} \cdot \frac{f_{\text{desmear}}\left(f_{\text{lin}}^{i}\left(S^{i}(x,y) - S_{\text{bias}}^{i}\right)\right) - S_{\text{dark}}^{i}(x,y)}{\Omega \cdot F^{i}(x,y) \cdot t_{\text{int}}^{i}}.$$
(10)

The retrieval process in MATS L1b is then simply described by equation (10). By unraveling this equation, we see that the following processing steps are necessary:

- 1. Subtract biases in the CCD readout electronics  $S_{\mathrm{bias}}^{i}$  (Section 5.3).
- 2. Correct for non-linearity effects by applying  $f_{\mathrm{lin}}^{i}$  (Section 5.4).
- 3. Desmear the images by applying the function  $f_{\text{desmear}}^i$  (Section 5.5).
- 4. Subtract the dark current of the CCD  $S_{\text{dark}}^{i}(x,y)$  (Section 5.6).
- 5. Divide by the flatfield,  $F^{i}(x,y)$ , of the particular channel (Section 5.7).

- 6. Calibrate the different channels with respect to each other by scaling them by relative calibration factors  $r^i$  (Section 5.8).
- 7. Apply the absolute calibration by multiplying with  $a/(\Omega \cdot t_{\text{int}}^i)$  (Section 5.8).

Figures 3 and 4 present examples of the image processing workflow for dayglow (when there is direct solar illumination of the airglow layer), and nightglow (when the airglow layer is not illuminated by the sun). These figures illustrate the impact of different substeps, providing a reference for the reader in the subsequent sections where these steps are detailed. Note that the index i, denoting the particular channel, will generally be omitted in the following sections, as procedures are performed on all channels.

#### 5.1 Single event detection




This section describes the single event detection, which, as described above, constitutes the first step of the preprocessing for the level 1a to 1b chain.

The image detector collects all electrons created in the sensitive volume, those originating from photons incident on the detector, thermally created ones (dark current), as well as those produced in direct ionisation by penetrating energetic particles. To reach the detector, particles must be capable of penetrating the structure between the detector and the open space. While the material thickness differs widely between the directions and individual detectors, it corresponds to at least a few millimetre equivalent Aluminium shielding, effectively blocking most electrons with energies up to a couple of MeV. However, protons with energies of multiple MeV can reach the detector.

The particle passing through the active volume is instantaneous compared to the exposure time, creating ionisation in the affected pixel. The ionisation may be created both by the primary particle, and the secondary particles produced by its interaction with matter. The enhanced ionisation is concentrated around the track of the primary particle. This effect of ionising radiation is known as Single Event effects, as opposed to cumulative damage due to ionising radiation (see Section 5.2).

The ionisation along the track of the ionising particle adds to the image. This results in clusters of pixels with enhanced counts. If the primary particle track passes perpendicularly to the detector, the cluster generally comprises a compact area a few pixels wide. For particles on grazing trajectories to the detector plane, the cluster can instead appear as a longer streak. Fig. 5 shows an example of a section of a full resolution image with one severe and multiple smaller single events. This specific image represents a worst-case scenario and was captured over the South Atlantic Anomaly, as discussed further below. In binned images, the clusters/streaks correspond to fewer pixels (combining a large number of unbinned pixels).

Originating from individual particles, pixel clusters with enhanced signal due to single event effects are confined to single images. This allows for easy detection by comparing with the temporally adjacent images. Thus, a routine for the detection of single event effects has been implemented, identifying the affected pixels as those where the difference from the average of the previous and subsequent images exceeds a threshold. Setting the threshold is a compromise between missing weaker single-event effects and misinterpreting the real image variations as single effects. A conservative approach is used, setting the threshold at five times the standard deviation of the image difference. For the IR1 channel this results in threshold values of less than 40 counts for nighttime measurements, and about 150-200 for normal daytime measurements. The detected single event

Figure 3. The left hand column shows an example of the different substeps of the calibration process for a dayglow image taken by the IR2 channel, demonstrating each calibration step (for consistency with figure 4 and so that the relative contribution of a correction process can be easily judged we show all steps despite similarities). The right hand column shows the corrections made by the process in question. a) The level 0 input image, b) The combined effect of bias, hot pixel and single event corrections in number of subtracted counts. The bias subtraction is a constant number over the entire image making the single events and hot pixel corrections stand out in the image. c) The image after single event and hot pixel correction and bias subtraction. d) The linearisation factor applied to each pixel. e) The image after linearisation has been applied. f) The number of counts subtracted to correct for smearing. g) The desmeared image. h) The dark current subtraction in counts. i) The image after dark current subtraction. j) The calibration factor  $(a \cdot r^i)/(\Omega \cdot F^i(x,y) \cdot t^i_{int})$  is applied. Since the relative and absolute calibration factors are constant for the whole image the structure in the field is solely due to the flatfield. k) The calibrated image.

**Figure 4.** Same as Figure 3 but for a nightglow example. As compared to the dayglow case, the bias subtraction has a larger relative effect on the signal strength, and the removal of hot pixel can be clearly seen in when comparing panel a and c. The low intensity of the nightglow signal means that it is highly linear and so that the linear correction factor de facto becomes 1 across the whole image, as can be seen in panel d.

**Figure 5.** Example of severe singular event impact on a section of an unbinned image, taken over the problematic region of South Atlantic anomaly (IR1 channel, 2023, August 30, 21:15:02 UT). The colour scale is in raw counts as registered by the detector (thus including bias, dark current etc.) and the axes show the unbinned column and row numbers.

pixels are flagged and their value is replaced by the median of the  $3 \times 3$  surrounding binned pixels, excluding any potential additional pixels that have been flagged as single events.

Fig. 6 presents the average number of pixels flagged as single event per image. For most locations, only a few single events are detected in a single exposure. The elevated occurrence of high-energy particles connected to the South Atlantic Anomaly (SAA) is clearly seen by the numerous single-event detections. In fact, the number of single events in an exposure can be so large that they dominate the image, increasing the standard deviation of the difference images to several hundred counts, which results in that only the most intense single event pixels are detected. The large number of single events makes the images acquired when the satellite is in the SAA very challenging to use for scientific analysis. Fig. 5 presents an image from the SAA region. Occurrence of single event is also enhanced in the subauroral region, seen as two bands in the 40° to 70° latitudes in both hemispheres.

#### 5.2 Hot pixel correction



Along with the immediate ionisation observable as single event effects, high-energy particles produce cumulative effects. The physical mechanisms of these effects are beyond the scope of this paper, but the direct consequence of the total dose effects is enhancement of the dark current level in some pixels. These enhancements are referred to as "hot pixels", and the removal of them constitutes the second step of the level 1a to 1b preprocessing. The level of hot pixel dark current is enhanced by tens to hundred counts per exposure in a binned pixel, and thus drastically exceeds the nominal dark current level. Hot pixels are created (or further enhanced) when high-energy particles impact the detector, and multiple such jumps of a pixel's dark current

**Figure 6.** Average number of single event affected pixels per image with count values exceeding 200, for IR1 channel between December 2022 and May 2023. Note that the marked position is the satellite position at the time the image was taken, not the tangent point of the measurement. This means that these single events will impact images of the atmosphere north or south of the marked region, depending on which way the satellite is moving (ascending or descending node).

appear in the SAA region. With time, the dark current in hot pixels typically decreases, supposedly due to annealing of the radiation-produced defects.

The number of hot pixels was negligible prior to launch, but has steadily increased with the satellite in orbit. After a month in orbit a significant fraction of unbinned pixels were affected. A dark full frame reference image provides a way to characterise the hot pixels. The corrections to the binned images can then be applied by binning the full frame dark images with appropriate binning parameters. However, this method has some operational implications. As the imagers are not equipped with shutters, the whole satellite needs to be pointed above the atmosphere to ensure a dark field of view. Nominal science operations are interrupted during this manoeuvre. Furthermore, full-frame uncompressed images are demanding in terms of telemetry bandwidth.



As only limited number of full-frame dark images have been acquired, and as the hot pixels are continuously created or annealed, a method to update the correction, based on regularly acquired binned images, is needed to give date-specific hot pixel correction arrays. Hence, a correction array (of the size of the binned image) is initialised from a full frame dark image, by applying the appropriate binning to the pixels. Only enhancements in excess of certain level (e.g. for IR1 channels chosen at 7 counts per binned pixel per exposure) are recorded, while enhancements of lower level are replaced by zeros. The correction

array is then subtracted from the subsequent images. The correction array is then updated by identifying pixels with a persistent offset from neighbouring pixel values. The underlying image is assumed to vary smoothly in the vertical direction, and a moving robust fitting window is used to quantify these persistent differences as the mean fitting error of the given pixel over a large number of images. These are used to adjust the hot pixel correction array. The hot pixel correction arrays are created for each day, taking the hot pixel correction of the previous day as a starting point for the procedure described above. They have been compared with the new full frame dark images with good results. Panel b of Fig. 3 and 4 illustrates the impact of correcting hot pixels and single events, as the adjusted pixels are distinctly noticeable. Hot pixel correction is not applied for the background channels (IR3 and IR4), where the effect, due to the heavy binning, is smaller and more difficult to correct for.

Comparing the hot pixel corrections between subsequent days gives an idea of the variability of those, and thus the error introduced by using a daily hot pixel correction. Analysis of IR1 hot pixel corrections for two weeks in March 2023 indicates that the average value of hot pixel correction increased by 0.28 counts/binned pixel per day during this period. The standard deviation of the day-on-day difference of hot pixel corrections is 4.6 counts/binned pixel. The standard deviation partly indicates the accuracy of determining the correction array in the aforementioned process, but it primarily represents the variability of the hot pixels. It is therefore used as an estimate of an error associated with the hot pixel correction, although it should be emphasised that this error is not fully random, the enhancement for a particular pixel during a short sequence of images will not vary from image to image with the above standard variation.

#### 5.3 Subtraction of readout electronics bias





An examination of (8) reveals that the first step of the main calibration process is to determine the electronic bias  $(S_{\rm bias})$ , so that it can be subtracted from S(x,y). The CCD contains 2048 columns used for imaging, with an additional 50 pixels on each side of the summation row that remain unlit during exposure and are cleared each time a row is read. These "leading/trailing blanks" allow for the determination of electronics bias. Tests have shown that the trailing blanks more accurately reflected the electronic bias than the leading blanks. Ideally, these 50 pixels should have the same value, excluding noise, but a slight gradient towards the illuminated pixels was observed. Hence, an average value  $S_{\rm bias}$  for the electronic readout bias is calculated using only 16 of the 50 trailing pixels. Although  $S_{\rm bias}$  can theoretically be measured for each read-out of rows, nearly identical values between rows justify the use of a single value for the entire image. The minimum value given in panel b of Fig. 3 and 4, reflects the magnitude of the bias correction, namely 292 and 290 counts, respectively.

#### 5.4 Correction for non-linearity and saturation effects

The second step of the main processing is to correct for non-linearity that the MATS readout chain exhibits when exposed to high signal levels, i.e. applying function  $f_{\text{lin}}^i$  in (10). To address this, pre-flight measurements were conducted to understand this non-linear behaviour. The characterisation of non-linearity involved measuring the dark current across all channels with progressively increasing binning factors until the ADC became saturated, so that the non-linearity function  $f_{\text{nonlin}}$  could be assessed (see Fig. 7).

**Figure 7.** Left: The non-linearity curve from the readout electronics fitted on data from the lab measurements (gray). The second order term in the non-linearity starts at 11 993 (dashed line) and the measured values above 32000 (dotted line) is marked to indicate ADC saturation. Right: Single pixel non-linearity data from IR (red) and UV (magenta) and the corresponding saturation levels (dashed curves).

A function




$$y(x) = \begin{cases} x, & \text{if } x \le e \\ b \cdot (x - e)^2 + x, & \text{if } x > e, \end{cases}$$

$$(11)$$

approximating  $f_{\text{nonlin}}$ , is then fitted to the data. The parameter x is here the value estimated from an unbinned reference measurement, multiplied by the binning factor (after correcting for bias), i.e. an estimation of the true value that should have been measured using an entirely linear sensor. The fitted coefficients e = 11993 and b = -0.000007 can then be used to represent and correct the in-orbit data for non-linearity by applying the inverse of the function of y, i.e.  $f_{\text{lin}}$  (see Eq. 10). The correction will not be perfect, and data with non-linearity correction larger than 5%, equivalent to a measured value of 25 897, are therefore flagged as "highly non-linear". This happens very rarely, in approximately in 0.3% of the images in the channels with the strongest signal (IR2). As a precaution, the data processing was made to flag any data above 32 000 counts as "saturated", but this did not ever occur in the current data set.

Saturation can also occur directly in the CCD. This happens when the number of electrons in an image pixel, shift register, or in the summation well, exceeds the capacity of that pixel/well. For the CCDs and binning factors used in the MATS instrument, only single-pixel saturation needs to be taken into consideration. To account for this, any data where the measured value, divided by the binning factor, exceeds an equivalent of 150 ke<sup>-</sup> is flagged as saturated. The saturation levels (after bias subtraction) become 4411 for the IR channels and 6617 for the UV channels. The difference in values is due to the different gains of the analogue amplifiers of the two channel types. Panel d of Fig. 3 and 4 shows the effect of the linearity corrections for day- and nightglow respectively. Due to the nightglow signal's low intensity, it exhibits strong linearity, resulting in a linear correction factor that effectively remains at 1 throughout the entire image.

## 360 5.5 Removal of readout smearing effect



The entire CCD is used to capture the image, and the rows of the image are sequentially transferred to the readout register and digitised during the readout process. Meanwhile, since the MATS instrument lacks a shutter, the rows awaiting readout are exposed to signal from a different part of the scene compared to their original position. This occurs for a small fraction of the nominal exposure time and adds a small additional signal. In the case of MATS, the readout is via the bottom of the image, which will be dark at night, but typically the brightest area of the CCD during the day. This makes the desmearing process much more important for daytime images.

The next step in the image processing chain is therefore to find the true image S from the readout-smeared image, which according to Eq. (10) is,

$$S_r = f_{\text{lin}}^i \left( S_c^i(x, y) - S_{\text{bias}}^i \right). \tag{12}$$

Assuming that the observed scene is constant during the exposure and read-out phases, the bottom row  $(r'_0)$  is read-out with no extra contamination, and i'th row  $(r'_i)$  is read-out as the sum of the original row  $r_i$  and a fraction  $t_R/t_E$  of the signal from the previous rows, where  $t_R$  is the read-out time and  $t_E$  is the exposure time, i.e.

$$r_i' = r_i + \frac{t_R}{t_E} \sum_{j=0}^{i-1} r_j.$$
 (13)

This can be expressed as a matrix equation

$$\mathbf{S}_{\mathrm{r}} = \left[ \mathbf{I} + \mathbf{L} \left( \frac{t_{\mathrm{R}}}{t_{\mathrm{E}}} \right) \right] \mathbf{S}.$$
 (14)

Here, **I** is the identity matrix and  $\mathbf{L}(x)$  is a lower triangular matrix with the value x at all positions below the main diagonal. The image  $\mathbf{S}_{r}$  is assumed to be in matrix format, with the bottom row of the image as the first row of the matrix. This means that the true image  $\mathbf{S}$  can be determined using the inverse equation

$$\mathbf{S} = \left[ \mathbf{I} + \mathbf{L} \left( \frac{t_{\mathrm{R}}}{t_{\mathrm{E}}} \right) \right]^{-1} \mathbf{S}_{\mathrm{r}}. \tag{15}$$

The equation can be solved using standard methods, and is what constitutes the  $f_{\text{desmear}}$  function in equation (8).

# 5.5.1 Handling cropped images

As mentioned above, to restrict the amount of data sent to the ground station, the images transmitted are limited to the relevant regions of the atmosphere mainly between 60 and 110 km altitude and are binned to different degrees depending on the need for spatial resolution. The background channels are binned to a greater extent than the main channels. The images may also be cropped to avoid areas affected by baffle vignetting and again to minimise data use. Cropping of the images at the top and

<sup>&</sup>lt;sup>1</sup>Note that the information between 60 km and 70 km is actually below the field of view, and is heavily affected by baffle interference. These rows are kept mainly because they are needed for the desmearing process.

sides has no affect on the de-smearing algorithm, but images cropped from below must be treated specifically. The cropping mechanism involves rapidly shifting (fast enough to introduce negligible smearing) the entire image down until the first row of interest is in the readout register. The smearing effect begins with at least part of the desired image in the region of the CCD that is exposed but not read out.

Consequently, we need to approximate the missing data in that region. This will involve extrapolation of the bottom rows of the part of the CCD actually digitised. In order to understand what this signal typically looks like, we have investigated data from all channels taken in the so-called full frame mode <sup>2</sup>. The uncropped vertical profiles of IR2 and UV2 can be seen in Fig. 8. It is evident that some form of extrapolation, either linear or exponential, should suffice in most scenarios. For nighttime images, a linear approach is likely the most suitable, although the low signal means the correction is not highly critical. During daytime, it is natural to expect the Rayleigh scattering to increase exponentially with decreasing altitude, which indeed provides the best fit. However, for UV2, the masked region is significantly larger and exhibits a peak in the signal due to atmospheric absorption. This requires a unique modelling approach. Through experimentation, we have determined that a Lorentzian curve effectively approximates the peak. During the night, when the signal can increase with height at the bottom of the image, a Lorentian cannot be fitted and no desmearing is done. This should not be a major problem, as low night signal results in very limited smearing in the first place, but the data is nonetheless flagged.

In the case of cropped images, the correction takes a different form. For the measured signal  $S_r$ 





$$\mathbf{S}_{r} = \mathbf{S} + \mathbf{L} \begin{pmatrix} t_{R} \\ t_{E} \end{pmatrix} \begin{pmatrix} F_{00} & \cdots & F_{0n} \\ \vdots & \vdots & \ddots & \vdots \\ F_{s0} & \cdots & F_{sn} \\ S_{(s+1)0} & \cdots & S_{(s+1)n} \\ \vdots & \vdots & \ddots & \vdots \\ S_{m0} & \cdots & S_{mn} \end{pmatrix} \quad \Rightarrow \quad \mathbf{S}_{r} - \mathbf{L} \begin{pmatrix} t_{R} \\ t_{E} \end{pmatrix} \begin{pmatrix} F_{00} & \cdots & F_{0n} \\ \vdots & \ddots & \vdots \\ F_{s0} & \cdots & F_{sn} \\ 0 & \cdots & 0 \\ \vdots & \vdots & \ddots & \vdots \\ 0 & \cdots & 0 \\ \vdots & \vdots & \ddots & \vdots \\ 0 & \cdots & 0 \\ 0 & \cdots & 0 \\ \vdots & \vdots & \ddots & \vdots \\ L(t_{R}/t_{E})_{(s+1)0} & \cdots & L(t_{R}/t_{E})_{(s+1)n} \\ \vdots & \vdots & \vdots & \vdots \\ L(t_{R}/t_{E})_{m0} & \cdots & L(t_{R}/t_{E})_{mn} \end{pmatrix} \mathbf{S}$$

$$(16)$$

which can again be solved for S by linear algebra. Here F and its components  $F_{ij}$  represent the part of the image ("fill") that is not considered in subsequent analysis. This image is assumed to have s rows.

Panel f of Figs. 3 and 4 shows the effect of desmearing, which can be significant in the upper part of the image, as this part has been exposed to light from the brighter lower atmosphere during read-out. The effect of smearing on the statistical uncertainty in the image can be calculated by multiplying the noise matrix by the weighting matrix. In addition, there is an extra systematic uncertainty introduced when cropped images are desmeared as we are forced to estimate the missing rows. The uncertainty is estimated by conducting the extrapolation separately using both a linear and an exponential function, and then considering half the difference between the results from the two methods as the uncertainty estimate. In most cases, this will at most introduce an error of 1-2 percent near top of the images and substantially less in the lower part of the image. However, this may be significantly larger in the case of UV2 where the cropped area is large.

<sup>&</sup>lt;sup>2</sup>This is a mode that takes full frame unbinned images and saves the entire un-cropped images. Due to the data limits, it cannot be run under normal operations.

**Figure 8.** Examples of un-cropped profiles for the IR2 channel and the UV2 channel, used for determining the best function to approximate the data in the cropped part of the image. The profiles of the IR2 channel show similar behaviour to those of the IR1, IR3, IR4 and UV1 channel (not shown). The profiles for UV2 on the other hand, where the CCD field of view is lower, are heavily influenced by ozone absorption and baffle vignetting at lower altitudes 2023-01-09.

#### 5.6 Subtraction of dark current



Studying equation (8), the next step is to subtract the CCD dark current  $S^i_{\rm dark}(x,y)$ , which is a function of temperature. In order to determine the relationship between dark current and temperature, the flight CCDs were placed in a dark cold chamber, where the dark current was recorded under varying thermal conditions. Figure 9 shows the average relationship obtained for one of the CCDs (IR1). It is evident that the dark current  $S_{\rm dark,e}$  in electrons per second follow the a log-linear relationship so that

$$\log(S_{\text{dark,e}}) = aT + b \tag{17}$$

where a and b are constants that are determined from dark chamber experiments.

The linear relationship was determined independently for each pixel. As hot pixels have already been compensated for in the previous step, they were removed by passing the coefficients a(x,y) and b(x,y) through a 3-pixel median row filter before calculating  $S_{\rm dark,e}(x,y)$ , and converting to  $S_{\rm dark}(x,y)$  in counts per second.

The temperature is determined using sensors placed on the CCD housing of the two UV CCDs. Therefore, the UV1 and UV2 temperatures correspond to the values recorded by their respective sensors, while the temperatures of the IR channels, which lack individual measurements, are set by averaging the readings from the two temperature sensors. Under normal conditions, the temperature of the CCDs remains below -10 °C, where the dark current is minimal. Consequently, the minor temperature variations among the CCDs have a negligible impact, and the overall effect of the dark current subtraction is marginal, ranging from 1% in UV1 to 0.01% in IR1. The minor effect of the dark current subtraction is evident by comparing the magnitudes of panels g and h in Figs. 3 and 4.

**Figure 9.** Average dark current in electrons per second, as a function of temperature as measured in the cold chamber for the IR1 channel CCD.

## 5.7 Flatfield correction


As mentioned above, the flatfield factor F(x,y) in equation (6) connects the reference area to all other pixels on the specific CCD. In other words, the sensitivity relative to the reference area  $A_{\rm ref}$  must be determined for each pixel. This process was carried out in the OHB clean room by configuring the instrument to capture an image of a uniformly white diffusive Lambertian screen, illuminated by the Xenon lamp. Ideally, flatfield measurements should have been taken in a dark room at low temperatures. However, the baffle, which greatly affects the flatfield edges, had not yet been installed during the Stockholm University dark room measurement campaign.

In order to remove the effect of the light from the surrounding room, each image of the screen was accompanied by a background image with the same exposure time, but with the Xenon lamp illuminating the screen switched off. In this way a residual image was constructed by subtracting the background images from the white-screen images. The residual images obtained in this way are free from readout bias and dark current, so that the signal strength is due only to incoming light. In addition, an external shutter was used to avoid smearing of the image. The saturation effects (see Section 5.4) were avoided by adjusting the brightness of the screen and the exposure time so that the images were acquired in the linear regime. The residual images were passed through a 2-dimensional 3-pixel median filter to remove hot pixels and single event occurrences, since these are compensated for separately (see Sections 5.2 and 5.1).

From the residual images the flatfield scaling factor F(x,y) for each pixel relative to the mean signal  $\bar{S}_{\gamma,A_{\text{ref}}}$  in the reference area  $A_{\text{ref}}$  of the particular channel was determined as

$$F(x,y) = S_{\gamma}(x,y)/\bar{S}_{\gamma,A_{\text{ref}}} \tag{18}$$

Panel j in Figs. 3 and 4 show the combined effect of the flatfield, the relative and absolute calibration). Note that the relative and absolute calibration factors are constant across a given image. Hence, the patterns seen in panels j represent the flatfield corrections.

Uncertainties may be introduced to this flatfield analysis by the use of a non-perfect (non-Lambertian) screen. In order to quantify this, we have measured the flatfield several times with different screen setups by rotating the screen and changing the distance between the lamp and the screen. The differences between these measurements provide an estimate of the systematic uncertainty of our flatfield corrections.

#### 5.8 Calibration






As a final step of the L1a to L1b processing, the measured signals need to be converted to the actual photon fluxes (radiances) received from the Earth's atmospheric limb. An absolute calibration of the limb instrument and a relative calibration between the spectral channels are required to determine the calibration factors  $a = a_{IR2}, a_{UV2}$  and  $r^i$  in equation 6. With regard to scientific data products, absolute calibration is the basis for a quantitative analysis of  $O_2$  airglow brightness and NLC scattering, which is needed for retrieving quantities such as atomic oxygen concentrations, ozone concentrations and the NLC ice water content. The relative calibration between the individual IR channels and UV channels, respectively, is decisive for the temperature retrievals of the  $O_2$  airglow emission and the microphysical particle retrievals from the NLC scattering. Regarding retrieval requirements for the different data products, an accuracy of  $\sim 10\%$  is required for absolute calibration, and an accuracy better than 1% is required for the relative calibration between channels.

In the following sections, we describe three approaches to absolute and relative calibration. Firstly, an instrument model describes the designed responses. Secondly, laboratory measurements carried out before launch provide the first experimental calibrations and a verification of the instrument performance with respect to the mission requirements. Thirdly, in orbit calibrations have been obtained by observation of stars. The instrument model and the lab calibration address the response of the instrument for a given radiance from an extended source, giving a measurement of  $a \cdot r^i/\Omega$  in 6. The star calibration, on the other hand, addresses the response of the instrument to the irradiance from a point source. Here, all calibration coefficients have been converted to the irradiance description of the star calibration. That is, the radiances received by each CCD pixel in the instrument model and the laboratory calibration have been multiplied by the solid angle  $\Omega$  that contributes to the illumination of that pixel, so that the factor  $a \cdot r^i$  is obtained (see equation 6). The solid angle is given by the CCD pixel size  $d = 13.5 \,\mu m$  and the effective focal length of the telescope  $f = 261 \, mm$ , as

$$\Omega = \frac{d^2}{f^2} = 2.676 \times 10^{-9} \,\text{sr}.\tag{19}$$

#### 5.8.1 Instrument model

An instrument model was developed based on the optical elements depicted in Fig. 1. This includes the telescope mirrors, the filters, beam splitters, and folding mirrors providing the spectral separation, as well as the CCDs. In the model, all optical elements are characterised by the optical properties provided by the manufactures (spectral reflectivities and transmissions,

and quantum efficiencies of CCDs). The responses of the CCD readout electronics (recorded counts per CCD electron) were measured as part of the instrument preparation. The resulting absolute and relative calibrations coefficients of the instrument design are summarised in Table 2.

Based on uncertainties of 2-3% for the reflectivity, transmission, or quantum efficiency of the individual components, the accuracy of this instrument model is 10%. This does not include possible long-term changes of optical elements e.g. due to contamination or ageing.

## 5.8.2 Laboratory calibration






Laboratory measurements were performed under two different sets of conditions. Prior to integration of the complete satellite, optical calibrations were performed on the instrument payload. These measurements were performed in a dark room at Stockholm University, with the payload placed in a cooling chamber, providing an instrument temperature down to approximately 0°C. After satellite integration, additional optical measurements were performed as part of general satellite testing at OHB Sweden. These measurements were limited to conditions with ambient illumination and room temperature. The calibration results reported in this section were obtained during the optimised conditions at Stockholm University.

The optical setup for the absolute calibration is identical to the setup for the flatfield calibration in Section 5.7. A 1000 W Xenon lamp illuminates a Teflon screen that fills the field of view of the instrument. In the spectral regions of interest, the Xenon lamp has been spectrally calibrated against reference light sources: a Cathodeon deuterium discharge lamp (200-400 nm) calibrated at the National Physical Laboratory (NPL) in the U.K., and a calibrated Oriel tungsten lamp (250-2400 nm) provided by Newport Corporation.

The absolute calibration focusses on the most sensitive channel in the two spectral regions, respectively, IR2 and UV2. The relative calibration then connects this absolute calibration to the other channels IR1, IR3, IR4 and UV1. Measurements performed by the MATS payload before satellite integration were conducted without the instrument baffle. Therefore, the absolute calibration measurements need to be restricted to a central part of the CCD area that is not affected by subsequent mounting of the baffle. To this end, the absolute calibration is reported as the integrated signal over the CCD pixels within the reference area  $A_{\rm ref}^i$ , which is defined as the CCD rows 351-401 (of 511) and columns 525-1524 (of 2048) for every channel. The absolute calibration factors for IR2 and UV2 obtained in this way are presented in row 2 ( "Laboratory Calibration") of Table 2.

The accuracy of these absolute laboratory calibrations is 15% and 30% for the infrared and ultraviolet channels, respectively. This accuracy is primarily limited by the absolute calibration of the Xenon lamp, its temporal stability, and deviations of the screen from a perfectly reflecting and Lambertian surface. In principle, better accuracy would be achievable by illuminating the Teflon screen directly with the calibrated tungsten and deuterium lamps. However, this was not done as the screen setup did not provide sufficient signal-to-noise ratio with the weaker deuterium lamp.

As for the relative calibration between the channels, the above setup is not suitable as the spectral features of the Xenon lamp (in UV and IR) and of the Teflon screen (in UV) are not known with sufficient accuracy. Rather, for the relative calibration, the well-defined deuterium lamp and tungsten lamp were used in a linear setup to shine directly into the limb instrument.

These lamps can be regarded as point sources and were placed at a distance of 1.4 m and 3.2 m from the instrument entrance, respectively. The resulting range of incident angles into the instrument leads to the illumination of a defined circle on each CCD. The relative calibration between the channels is then obtained as the ratio between the integrated signals in these circles. In the image centre, the resulting ratio in the infrared is 3.396:1:5.970:8.599 for IR1:IR2:IR3:IR4. The resulting ratio in the ultraviolet is 2.451:1 for UV1:UV2. Combining the absolute calibration coefficients for IR2 and UV2 with these relative calibration factors gives the calibration factors reported as "Laboratory calibration" in Table 2.

The accuracy of these relative measurements is 1% and 3% for the infrared and ultraviolet channels, respectively. This accuracy is largely limited by the knowledge of the lamp spectra. However, an additional uncertainty is introduced by the fact that the above setup with a point source only utilises a subset of all optical paths that a distant atmospheric source would utilise through the instrument. Any inhomogeneity in the instrument optics will thus limit the validity of the above ratios as a relative calibration between the instrument channels.

#### 5.8.3 Star calibration






As explained above, stars are natural point sources and enable in-flight calibration of the imaging system. During routine operations, with MATS pointing to the limb and taking binned images, stars are repeatedly seen crossing the field of view. As these opportunistic star sightings are numerous, they can be used for absolute calibration as explained below.

With the MATS telescope pointed to the limb and close to the orbital plane, the optical axis of the telescope makes one complete rotation in one orbital period. The field of view covers a 5.67° wide swath of celestial sphere, defined by the orbital plane. As MATS' orbit is sun-synchronous, the orbital plane completes one full rotation in one year, thus covering almost the entire celestial sphere (apart from a few degrees wide regions around the north and south pole, as the orbit inclination is about 98°). Any star within the covered sky is observed twice per year for a period of a few days, as the orbital plane rotates past the star.

As MATS moves along the orbit, the stars are seen rising above the horizon in the field of view, moving at the apparent angular rate corresponding to the orbital rate. With integration times of a few seconds, stars are seen as vertical streaks in the images. In addition to the star, the images contain foreground emissions (airglow, atmospheric scattering), as well as possible artefacts such as straylight, hot pixels, single event effects. For the star calibration, the star signal thus must be separated from these other signals.

In our analysis, this is done by taking the difference between two subsequent images. The hot pixel pattern is essentially the same between the images (hot pixels develop and anneal on time scales long compared to the image cadence). Straylight, atmospheric scattering, and foreground emissions do vary between subsequent images, but the variation is in most cases small. In a second step, subtracting the average of two columns adjacent to the star streak removes the remaining difference in the non-star emission, as the foreground signal often is smooth in the horizontal direction. However, any single event effects (in either the current or preceding image) are not removed by this procedure.

In the further analysis, only star streaks confined to one binned pixel column are considered. For those, the total excess counts (from the differential image) associated with the star streak are integrated over the whole streak. The counts are low

enough to be well within the linear response region of the CCD. The total counts obtained in this manner are proportional to the total photon flux into the instrument. The absolute calibration coefficients  $a \cdot r^i$  (in photons cm<sup>-2</sup>nm<sup>-1</sup>count<sup>-1</sup>) in equation 6 can thus be determined by relating the observed signal (in counts s<sup>-1</sup>) to the tabulated star spectra from star catalogues (in photons cm<sup>-2</sup>s<sup>-1</sup>nm<sup>-1</sup>). The six MATS limb channels are equipped with narrow passband filters. For the calibration, we relate the spectral fluxes at the centre of the passband to the observed fluxes. This assumes that stellar spectra are smooth inside the passband. This assumption is justified as the spectra of the stars used in the calibration lack sharp spectral features. The four infrared channels are sensitive to late spectral class stars, while the UV channels only detect bright early spectral class stars.

The absolute calibration factors  $(a \cdot r^i)$  were obtained by associating the measured signal with the tabulated spectra for each star, obtained from Alekseeva et al. (1996); Pickles (1998) and Rieke et al. (2008). The overall estimation for a specific channel was determined as the mean of a Gaussian distribution that was fitted to all star observations for that channel. The error estimate is based on standard deviation of the distribution, and not the standard deviation of the mean. This is justified as the variance in the results clearly suggests that the individual star calibrations cannot be treated as independent measurements of the same quantity and, thus, the standard deviation of the mean would result in a too small error estimate. The resulting calibration factors for the different channels are given Table 2. The uncertainties are estimated to 3% for channels IR1 and IR2, and 5-10% for the remaining channels.

For the relative calibration, a slightly different approach was used, to avoid the systematic uncertainties due to the uncertainties of the absolute values or the tabulated star spectra. To this end, the absolute calibration coefficients were determined separately for each star and each channel. These coefficients of a particular channel were then plotted against the equivalent coefficients of the reference channel (see Fig. 10 for the relative calibration between channels IR1 and IR2), and the slope of the fitted line provided the relative calibration factor between these channels, with associated error estimate.

#### 5.8.4 Calibration summary








Table 2 summarises the above calibration results from the instrument model, the pre-flight laboratory calibration and the inorbit star calibration. As described in Section 5.8.2, laboratory activities were divided into absolute and relative calibration. The corresponding row in Table 2 is based on the absolute calibration of channels IR2 and UV2, and the relative calibration to infer the calibration coefficients for channels IR1, IR3, IR4, and UV1.

Following the discussions in the previous sections, the accuracy of the calibration coefficients in Table 2 is 10% for the instrument model, 15% (IR) respectively 30% (UV) for the laboratory calibration, and 3 to 10% for the star calibration. In the infrared channels, the calibration coefficients are consistent within these uncertainties. In the two ultraviolet channels, a decrease in sensitivity is apparent over time from the component-based instrument model via the laboratory calibration to the in-orbit star calibration. This is particularly evident at the shortest wavelength (UV1,  $\lambda = 270 \,\mathrm{nm}$ ) and is attributed to contamination and ageing of optical components. We have investigated a possible in-flight degradation during the data set, and there is indeed an indication of an increasing trend of around 10% in the absolute calibration coefficients for the UV channels. However, the limited time period (only two months of UV data) prevents us from determining the trend with sufficient certainty

**Figure 10.** Example of how the relative calibration coefficients, in this case IR1 versus IR2, were determined. Each point marks the absolute calibration coefficients obtained for the two channels for one particular star. The relative calibration factor is given by the slope of the fitted line.

to correct for it. Instead, the effect of any degradation trend is implicitly included in the estimate of the absolute calibration uncertainty. It should be noted that this degradation only affects the absolute UV calibration and that there is no indication of trend in the IR channels or in the relative calibration coefficients in either UV or IR.



The differences in relative sensitivity between the laboratory calibration and the stellar measurements are difficult to reconcile. Particularly, the relationship between IR3 and IR4. To investigate which was more correct we have utilised independent data. Firstly, we have images of the moon taken with all channels. These observations strongly support the stellar data. Secondly, comparisons of the Rayleigh scattering signal in the background channels IR3 and IR4 also strongly indicate that the stellar relative calibration is the more trustworthy. Hence, the coefficients based on star calibration as given in last row in Table 2 are used in the MATS level 1b data product (v 1.0).

| Calibration coefficients                                         | IR1             | IR2             | IR3             | IR4             | UV1            | UV2       |
|------------------------------------------------------------------|-----------------|-----------------|-----------------|-----------------|----------------|-----------|
| Instrument model [ph cm <sup>-2</sup> nm <sup>-1</sup> / counts] | 7.95            | 2.66            | 19.2            | 24.0            | 16.4           | 16.0      |
| Laboratory calibration [ph cm $^{-2}$ nm $^{-1}$ / counts]       | 11.3            | 3.32            | 19.8            | 28.6            | 41.7           | 17.1      |
| Star calibration [ph cm $^{-2}$ nm $^{-1}$ / counts]             | $10.0 \pm 0.5$  | $2.97 \pm 0.09$ | 21.5± 1.4       | $26.6 \pm 2.2$  | 54.3± 4.3      | 21.7± 1.6 |
| Relative calibration from stars                                  | $3.35 \pm 0.01$ | -               | $7.12 \pm 0.06$ | $8.99 \pm 0.06$ | $12.50\pm0.02$ | -         |
| Combined [ph cm <sup>-2</sup> nm <sup>-1</sup> / counts]         | 9.94± 0.34      | 2.97± 0.09      | 21.1± 0.8       | 26.7± 1.0       | 54.1± 4.4      | 21.7± 1.6 |

**Table 2.** Summary of the MATS calibration coefficients  $(a \cdot r^i)$  obtained using the different methods of calibration as explained in Sections 5.8.1 - 5.8.3. The combined absolute and relative calibration coefficients from stars (last row) is used in the level 1b data product (v 1.0).

## 5.8.5 Temperature dependence







A certain sensitivity of the signal to the temperature of the instrument is expected due to the temperature sensitivities of the CCDs, of the CCD readout chain, and of the interference filters. This can affect both the absolute and the relative signals.

The narrowband filters used in MATS were produced by Omega Optical Inc., where one of the key selection criteria was their limited sensitivity to temperature variations. The temperature dependence can be assumed to be described by a linear shift of the centre wavelength of the filter curves. The specified linear dependence of the filters was  $0.002 \text{ nm K}^{-1}$ . Measurements of the filter transmission curves in the darkroom at  $+22^{\circ}\text{C}$ ,  $0^{\circ}\text{C}$ , and  $-25^{\circ}\text{C}$  suggest a coefficient of  $0.007 \text{ nm K}^{-1}$ . This is larger than specified by the manufacturer, but still small enough to have a negligible effect.

To investigate the absolute sensitivity of the instrument to temperature, tests were performed with the instrument in a cooling chamber, using the same setup as for the absolute calibration (see Section 5.8.2), while varying the temperature from  $2^{\circ}$ C to  $20^{\circ}$ C. A quasi-linear temperature dependence of 0.1% K $^{-1}$  to 0.3% K $^{-1}$  was found. The temperature dependence is likely due to thermally-induced increased clock-induced charge, which arises from charge packets created during the transfer of charges between pixels and becomes more significant as the temperature increases. Since the CCDs are operated at much lower temperature (-20 to - $10^{\circ}$ C) than could be achieved in the laboratory, the measured temperature dependence can be considered an upper limit. During normal operation in orbit, the instrument is very stable in temperature and varies less than  $1^{\circ}$ C on a daily basis. This results in an absolute signal change due to the temperature of less than 0.4% and a relative change between IR1 and IR2 of around 0.2%. However, seasonal changes in the solar illumination of the satellite lead to overall temperature variations of approximately 5  $^{\circ}$ C. Since the star observations, on which the absolute calibration is based, have been performed during the whole dataset, any impact of the temperature on the signal would have been randomly sampled, and as such included in the star calibration error estimate.

#### 5.9 Example of calibrated images

Fig. 11 shows an example of calibrated images for the six channels of the limb instrument for a case where both airglow and NLC were observed.

In the IR1 and IR2 channels, the broader and less intense airglow signal is overlayed by the sharper and more intense signal from the NLC. The background channels, IR3 and IR4, evidently do not contain the airglow signal, and the coarser binning is apparent. In the IR3 image, slight straylight can be seen in the upper left corner, and also the enhanced signal in the middle portion might be related to straylight. As mentioned earlier, straylight will be addressed in the next data level. The UV channels cover a smaller altitude range, so the NLC layer appears higher in the image. As Rayleigh/Mie scattering is stronger in shorter wavelengths, fainter NLC structures below the tangent point peak are seen in the UV channels but not in the IR channels. Rayleigh scattering from the lower denser atmosphere is observed in all channels but is less prominent in UV1 because of the strong ozone absorption in this wavelength. It is also clear that hot pixels remain a challenge in the UV1 channel.

**Figure 11.** Example of calibrated images for an observation of dayglow and NLC, for the different limb channels. The white lines mark the altitude in metres.

## 6 Effects not compensated in the L1b data products

Here we list the known effects that are not compensated for in the L1b data product. They are either considered small (see estimates below), or they are easier to compensate for in the higher-level data products, such as tomographic temperature retrieval.

#### 6.1 Ghosting



Ghosting refers to the appearance of unintended secondary images, typically faint and blurred, adjacent to or superimposed upon the primary image of interest. This phenomenon arises because of reflections or scattering of light within optical components, such as lenses or mirrors. In principle, ghosting can be corrected for to some extent if the original image is in the field of view of the camera. However, this is often not the case. The ghosting in the different channels was characterised prior to launch. For the UV-channels the main ghosts appear approximately 150 unbinned pixels below and above the main image and were measured to about 1% (of the signal of the main image) in the UV1 channel and 5% in the UV2 channel. Due to saturation of the real image during the laboratory measurements, these values are likely overestimated. For the IR channels, weak ghost images of less than 1% were observed above and to the side of the original image.

## 6.2 Stray light

Stray light in the instrument is mainly due to illumination of the inside of the baffle by the lower atmosphere. For UV1 and UV2 the stray light is generally very small as stratospheric ozone effectively absorbs the upwelling radiation. Considerable stray light on the order of  $1 \cdot 10^{14}$  photons m<sup>-2</sup>sr<sup>-1</sup>s<sup>-1</sup>nm<sup>-1</sup> is seen for the two background channels IR3 and IR4, increasing slowly towards lower altitudes. Straylight is expected to be lower in the two A-band channels IR1 and IR2 due to partial absorption of the upwelling radiation by  $O_2$ . Stray light effects will be handled in the Level 1c data product. In Level 1c, the data of the individual channels will be combined, and data from the background channels IR3 and IR4 and the nadir photometers can be used to quantify stray light contributions.

#### 6.3 Polarisation

Due to the use of mirrors and beam splitters in the optical setup, the instrument exhibits sensitivity to the polarisation of the incoming light. The polarisation sensitivity has been measured with a linear setup consisting of a light source, a linear polariser, and the instrument. Sensitivities can be expressed as the amplitude of the signal variation over a full rotation of the polariser. The resulting sensitivities are  $\sim 5\%$  for the channels IR1, IR3 and UV1, and  $\sim 15\%$  for channels IR2, IR4 and UV2. Comparing to Fig. 1, a polarisation sensitivity of  $\sim 5\%$  is found for channels comprising one beam splitter or folding mirror, while a polarisation sensitivity of  $\sim 15\%$  is found for channels comprising two or more beam splitters or folding mirrors.

All calibration results presented in the current paper are based on unpolarised light. For much of the targeted MATS research, analysis of MATS atmospheric measurements, polarisation sensitivity is not critical. Airglow as major measurement target in the IR is unpolarised and thus not affected. NLCs as major measurement target in the UV can polarise scattered light in complex

ways depending on the shape and the size distribution of cloud particles. As usual for satellite observations of NLCs, Level-2 retrievals of cloud properties need to start out from pre-defined assumptions on particle shape and size distribution. Based on these assumptions, scattering simulations, including polarisation, are then performed to generate look-up tables connecting optical measurements and cloud properties (e.g. Hultgren et al., 2013). Molecular Rayleigh scattering as the major (daytime) background is strongly polarised, as scattering angles around 90° dominate the scattering in the MATS field of view, both for direct sunlight and for radiation upwelling from lower altitudes. Variations in measurement geometry along the orbit result in variations of polarisation and, thus, in a variation of the detected Rayleigh background. This variation is generally less than the variability of the Rayleigh background due to varying atmospheric density. It will be removed as part of the Rayleigh background subtraction in the MATS Level-2 processing.

## 7 Uncertainty estimation and error handling




As part of the Level 1b data product, a function for calculating approximate uncertainty of each pixel of the image is provided. The function returns an estimate of the statistical (random) uncertainty and the systematic uncertainty. Every level 1b image is also accompanied by an processing error flag matrix of identical dimensions, which highlights any anomalies that occurred during the calibration process. Inexperienced users are advised to use only pixels that are not flagged.

# 7.1 Estimation of the statistical uncertainty

**Figure 12.** Mean statistical uncertainty in counts for various error types for the IR1 channel, presented for dayglow (left) and nightglow (right).

Statistical errors are errors which are quasi-random or highly variable across the data. For the MATS 11b data, the errors that were considered in the error analysis are:

#### 1. Shot noise in the CCD detector.

- 2. Readout noise from readout electronics.
- 3. Digitization noise from the ADC.
- 4. Compression noise from the JPEG compression on board.
- 5. Errors from uncertainty in the hot pixel and single event correction.
  - 6. Noise propagating through the desmearing correction

Shot noise is calculated from the total signal recorded on each binned CCD pixel. This includes signal from incoming photons, both during exposure and readout (smearing), as well as dark current signal (incl. hot pixels). The standard deviation of the shot noise in electrons  $\epsilon_{Se}(x,y)$  is given by the square root of electrons in a binned pixel x,y

$$\epsilon_{Se}(x,y) = \sqrt{S_r \cdot \alpha / C_{\rm amp}},$$
 (20)

which when converted back into counts results in



$$\epsilon_S(x,y) = \epsilon_{Se}(x,y) \cdot C_{\rm amp}/\alpha = \sqrt{S_r \cdot C_{\rm amp}/\alpha}$$
 (21)

where  $S_r$  is the image read out from the CCD given by equation 12, and  $\alpha$  is the gain expressed in units of electrons per count.  $C_{\text{amp}}$  is a hardware-implemented amplification factor used to enhance the sensitivities of the UV channels, and equals 1 for the IR channels and  $\frac{3}{2}$  for the UV channels.

Note that the signal  $S_r$  includes shot noise contribution from photons hitting the CCD during exposure, photons hitting a pixel during readout (smearing) as well as dark current. Thus, even if the smeared and dark current signal is correctly compensated for, there will be residual shot noise in the final calibrated images from these two sources.

Readout noise is noise from the readout electronics, this value is based on pre-flight measurements of each channel where 690 the noise at "zero signal" is estimated from measured dark current data.

Digitisation noise is the uncertainty from the inherent limitation of the digital resolution. Since MATS uses 12 bit images created by truncating 16-bit data from the ADC, the digitisation noise (in 16-bit counts) varies depending on the bit-truncation (window-mode) used. For nightglow, when the signal is low, only the 12 least-significant bits are used, and digitisation noise corresponds to 1 count. For dayglow, it can vary, with a typical value of 4-8 counts.

After the image is truncated to 12 bits it is compressed to a JPEG image. The jpeg compression quality is adjustable, but generally it is set to 95%. Comparisons based on comparing uncompressed and compressed simulated data indicates that this results in an average uncertainty across pixels of 3.3 counts (least significant bits) in the 12 bit image. In the same way as the digitisation error, the JPEG compression error in 16-bit counts will depend directly on the bit-window used for the measurement. For nightglow, the resulting uncertainty will typically be 3.3 counts, while for dayglow it is around 13-27 counts. Also, note that while we consider the jpeg error as random in our model, it is not entirely random in practice, as jpeg artefacts will tend to manifest themselves as chequerboard patterns around strong gradients in the image (caused by e.g. single events).

Thus, depending on how the MATS data is used, one might have to take special care not to misinterpret any such pattern seen in the data.

As explained in section 5.2 hot pixel error arises from the fact that there is a slight variability in the hot-pixels in time. This means that they cannot be completely corrected for. Uncertainty from this error is estimated to 5 counts from looking at the daily variability of the hot-pixel maps. The hot-pixel pattern (or residual pattern for IR1 and IR2) will be uncorrelated across pixel, but highly correlated in time for a single pixel. As such, it might be considered a statistical or systematic error, or it might even be corrected for at a later processing stage, depending on the final usage of the MATS data. For single events, no direct uncertainty estimate is made since the error is highly localised in time and pixel position, and thus uncorrected single events would be considered outliers in the dataset, rather than an error to be characterised with a certain distribution.


**Figure 13.** Typical statistical uncertainty in relative units. Left: uncertainty during daytime, comprising dayglow in the IR and NLC in the UV. Right: uncertainty during nighttime, comprising nightglow in the IR and no NLC in the UV.

The last statistical error considered is the error that arises directly from the desmearing estimation. Since desmearing signals from pixels within the same column are interconnected, and these signals contain noise, there is a small secondary effect from uncertainty in this estimation. However, through modelling it was found that this effect is negligible and will not be considered for further analysis.

The total uncertainty is estimated by propagating each uncertainty estimate independently through the processing chain to either  $S^i_{\gamma}$  (Eq. 8) for uncertainty in counts, or all the way to  $L^i(x,y)$  (Eq. 7) for expressing the uncertainty in radiance. Then, assuming that the errors are independent, the total statistical uncertainty  $\varepsilon^i_{L,\,\mathrm{stat}}(X,Y)$  can be calculated as the root sum square of the different uncertainties.

Typical results for day and nightglow in the IR1 channel are shown in figure 12. For dayglow the overall signal is strong and the noise is dominated by shot and compression noise, while during nightglow conditions, when the overall signal is weak, the uncertainty in the hot pixel correction becomes dominating. This error will manifest itself as a highly time-correlated random pattern of dots in the images.

The relative noise, i.e. the total noise divided by the calibrated image for the different channels is shown in figure 13. For dayglow, the statistical uncertainty is below 1% for most of the image, increasing to about 3% at the top. However, for nightglow, the relative uncertainty is much higher. For the IR1 and IR2 it exceeds 5% above 100 km, while the background channels show a more moderate increase from 2-8 % going from 60-110 km.

For the UV channels the statistical uncertainty varies quite drastically (in relative terms) across the relevant altitudes. For the data where NLC are present, the error is 2 and 1% of typical NLC radiance for UV1 and UV2 respectively.

## 7.2 Estimation of the systematic uncertainty



730 In addition to random or highly variable errors, systematic errors (biases) may result from uncertainties in the calibration of the instrument and assumptions made in the data processing. These identified uncertainties are

- 1. Uncertainty in the dark current characterization of the CCDs
- 2. Uncertainties in the bias of the readout electronics
- 3. Uncertainties in the non-linear correction
- 4. Uncertainty of the background extrapolation used in the desmearing algorithm
  - 5. Uncertainty in the flat-field characterization
  - 6. Uncertainty in the absolute and relative calibration

The dark current estimation is based on pre-flight measurements and as such there is an uncertainty in its validity, both due to changes in the CCD itself, as well as uncertainty in estimation of the CCD temperature. However, since the dark current itself is very small at the operating temperatures of MATS CCDs (-15°C to -20°C), the secondary effect of the error in the estimation of the dark current becomes negligible.

The readout electronics bias is determined as the mean value of a set of blanks from the readout register, as described in section 5.3 Subtracting the wrong bias could lead to large relative errors where the signal is weak (e.g. above the airglow layer). The uncertainty in the bias is estimated by calibrating images using the trailing and leading blanks, respectively, and taking the half of the difference of the images as the uncertainty.

The non-linear correction (section 5.4) is estimated from a limited set of lab measurements. Furthermore, since many pixels are read out separately and added together to "simulate" a binned image any error in read-out bias estimation can impact the non-linearity curves. In order to estimate the uncertainty we assume that the applied correction only corrects for 50% of the non-linear effect. This value is based on variances in the non-linearity estimate seen between the range of different methods tested out for the correction, and is judged conservative.

For the desmearing algorithm to work, the lower atmosphere needs to be estimated. As a pragmatic way to estimate the uncertainty of the procedure, images were calibrated using both exponential and linear extrapolation of the lowermost pixels, as explained in 5.5. The  $1-\sigma$  uncertainty was then estimated as half of the difference between the resulting images.

Figure 14. Mean systematic uncertainty for various error types for the IR1 channel, presented for dayglow (left) and nightglow (right).

The uncertainties in the flat-field calibration and absolute calibrations are determined as described in Section 5.7, and the absolute calibration uncertainties are taken from Table 2.

The estimated uncertainties from each of the sources mentioned above are propagated through the calibration chain to give an estimated uncertainty in the calibrated radiances. Assuming that these errors are unrelated, we calculate the combined systematic uncertainty  $\varepsilon^i_{L_i, \text{SVS}}(X, Y)$  of channel i as

$$\varepsilon_{L,\,\rm sys}^{i}(X,Y) = \sqrt{\varepsilon_{\rm bias}^{i}(X,Y)^{2} + \varepsilon_{\rm lin}^{i}(X,Y)^{2} + \varepsilon_{\rm desmear}^{i}(X,Y)^{2} + \varepsilon_{\rm dark}^{i}(X,Y)^{2} + \varepsilon_{\rm flatfield}^{i}(X,Y)^{2} + (\varepsilon_{\rm relative}^{i})^{2} + (\varepsilon_{\rm absolute})^{2}}.$$
(22)

Comparing the different contributions to the overall systematic uncertainty in Fig. 14, the uncertainty of the absolute calibration  $\varepsilon_{\rm absolute}$  dominates in most circumstances, except for high altitudes during nightglow, where uncertainties in the bias subtraction could lead to significant errors.

Comparing the total uncertainty for each channel to the measured signal strength (Fig. 15), we find that the systematic uncertainties generally are around 3% for dayglow for all IR channels (with IR3 having a higher uncertainty than the rest). For nightglow it is closer to 3-4 % of the signal, except at higher altitudes where the uncertainty in the bias correction starts contributing leading to a systematic uncertainty of over 10% of the signal strength over 102 km.

For the UV channels the systematic uncertainty is significantly larger. For typical NLC data the uncertainty is around 10% up to the NLC layer then increasing above it. For background UV data the signal is very weak, thus the systematic uncertainty becomes large, up to 100% for UV2 at higher altitudes.

Overall, MATS images are dominated by the uncertainty in the absolute calibration. This is decisive for the accuracy of retrieving absolute quantities from the radiative measurements. In the IR, this includes airglow radiances, volume emission rates, and related retrievals of densities of atomic oxygen or ozone. In the UV, this includes retrievals of NLC volume scattering coefficients or ice water content. On the other hand,  $\varepsilon_{absolute}$  does not affect the retrieval of data products based on the relative

**Figure 15.** Typical systematic uncertainty in relative units. Left: uncertainty during daytime, comprising dayglow in the IR and NLC in the UV. Right: uncertainty during nighttime, comprising nightglow in the IR and no NLC in the UV.

signal between channels. This includes the derivation of temperatures in the IR and the derivation of NLC particle sizes in the UV. An error analysis of these quantities must use a modified equation 22 as input, with  $\varepsilon_{\rm absolute}$  removed.

#### 8 Summary and Conclusions

790

The MATS satellite's mission is to map gravity waves on a global scale in three dimensions and analyse the waves' characteristics. This is accomplished through tomographic analysis of images of wave patterns in airglow or NLC, captured using a limb-viewing telescope. These images serve as input to the tomographic retrieval, but can also be used for independent studies of, for example, large-scale patterns. This paper has outlined the so-called Level 1b data product, which is now available to the public for data collected between February and May 2023. In May 2023, the satellite's reaction wheels began to malfunction. A new steering algorithm using magneto-torquers is currently under development, though its effectiveness and the potential to continue the dataset remain uncertain.

In this paper, we have described the data processing (version 1.0) that converts raw MATS limb images into geolocated and calibrated images with tangent altitude information. This includes the geolocation technique and the various image processing steps, which serve to remove or compensate for instrumental effects. We have further detailed the calibration process, where the CCD pixel counts are converted into the radiance of the imaged limb.

Three different calibration techniques were employed: calibration based on the manufacturer's specifications for the throughput of optical components, laboratory calibration, and in-orbit calibration using the known brightness of stars. For the infrared channels, the absolute calibration coefficients obtained through the three calibration methods differed by no more than 20%, which is acceptable given the uncertainties of the approaches. In the ultraviolet range, both channels show a noticeable decline in sensitivity over time, from the component-based instrument model, through laboratory calibration, and to the in-orbit star calibration. As the strongest effect is seen at the shortest wavelengths, the reduction is likely a sign of contamination or ageing

of the optical components. Among the different calibration methods, the calibration based on stars had the smallest absolute uncertainty (between 3% and 10%), and thus these coefficients are used in the L1b calibration (as of version 1.0). During daytime, calibration uncertainty is the main contributor to the absolute error. However, at night and at low signal levels, other error sources become prominent, and the total error can, for instance, in the case of high altitude nightglow, reach 20%. The error can thus vary substantially with time and across the image, and it is recommended to use the error estimate function that is provided with the Level 1b data product.

795

Absolute calibration is the basis for a quantitative examination of O<sub>2</sub> airglow intensity and NLC scattering, and as such is used to determine parameters such as atomic oxygen levels, ozone concentrations, and ice water content in NLCs. The relative calibration between the individual IR channels and UV channels, on the other hand, is important for temperature and NLC particle size retrievals, imposing stricter accuracy requirements on the relative calibration than on the absolute calibration. The star calibration provides an absolute accuracy of 3% to 10% depending on channel, and around 1% for the relative calibration between the IR channels and the UV channels, respectively.

Code and data availability. The MATS level 1b dataset can be accessed from the Bolin centre database at https://doi.org/10.17043/mats-level-1b-limb-cropd-1.0. The code used to produce the level 1a and level 1b datasets is available on github repositories https://github.com/innosat-mats/level1a and https://github.com/innosat-mats/MATS-L1-processing, respectively.

Author contributions. The MATS science mission was conceptually developed, and its instruments were designed by JG, LM, OMC, DPM, NI, JH, JD and JS. Laboratory testing and calibration measurements were performed by LM, JG, OMC, BL, DPM, NI, JH, JD, GG, GO, and JS. The final launch site tests were completed by LM, BL, and JD. The retrieval and calibration analysis was developed by LM, JG, OMC, BL, DPM, NI, JH, JD, LK, and JS. The manuscript was primarily written by LM, DPM, OMC, NI and JG, with contributions and comments from the other coauthors.

Competing interests. JG is a member of the editorial board of Atmospheric Measurement Techniques.

Acknowledgements. This work has been financed by the Swedish National Space Agency under the grants 21/15, 297/17, 2021-04876, 2022-00108 (Stockholm University), 22/15, 298/17 (Royal Institute of Technology) and 23/15, 299/17 (Chalmers University of Technology). The authors thank Omnisys Instruments, OHB Sweden and Rocket Lab New Zealand, for support and use of premises during the calibration campaigns. AI coding and text editing assistance has been used.

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
