# Peer review of "The MATS satellite: Limb image data processing and calibration"

_EGUsphere, 2025_

## Author Response (AR1)

We thank the reviewers for their time and for useful comments, which we address below (original comment in *black italics*).

**RC1 (Anonymous Referee #2, 11 Apr 2025)**

Citation: https://doi.org/10.5194/egusphere-2025-265-RC1

This article deals with the data processing and calibration of MATS satellite images. This is important for assessing data quality in scientific studies such as gravity waves and NLC characteristics in the mesosphere and lower thermosphere (M-LT). Gravity waves are an important component of the dynamics of the M-LT region and a good description of their three-dimensional structure is necessary to better understand their role and the mechanisms involved.

The description of all the data processing phases is very detailed. All the error budget terms are carefully estimated. This document could be an interesting contribution to AMT. However, it lacks a section showing some examples of scientific results to really assess whether the quality of the processed and calibrated data is sufficient to meet the main scientific objectives of the MATS mission on gravity waves and NLCs. Yes, in some way it would be instructive to show scientific results. However, this is beyond

Yes, in some way it would be instructive to show scientific results. However, this is beyond the scope of this AMT manuscript, both in terms of length and in terms of content. This manuscript is based on pre-launch activities (except for the star calibration), and it is supposed to be the major reference about the MATS calibration for all future publications of scientific results. So we want people to refer to this manuscript. But we do not want people to refer to this manuscript when it comes to the tomography retrieval and to first scientific results from MATS. First science results will be published in different papers, and these papers will need a lot of additional description and explanations about methodology and processing. Some such papers are currently underway (see references below) with different lead authors. These papers, not the current calibration manuscript, should be the original publications of the science results in question.

Linder, B., Gumbel, J., Murtagh, D. P., Megner, L., Krasauskas, L., Degenstein, D., Christensen, O. M., and Ivchenko, N.: Joint observations of oxygen atmospheric band emissions using OSIRIS and the MATS satellite, EGUsphere [preprint], https://doi.org/10.5194/egusphere-2025-493, 2025.

Linder, B., Krasauskas, L., Megner, L., and Murtagh, D. P.: The global O2 airglow field as seen by the MATS satellite: strong equatorial maximum and planetary wave influence, EGUsphere [preprint], https://doi.org/10.5194/egusphere-2025-1470, 2025.

Pérez-Coll Jiménez, J., Ivchenko, N., Lindstein, C., Krasauskas, L., Hedin, J., Murtagh, D. P., Megner, L., Linder, B., and Gumbel, J.: A statistical study of the O2Atm(0-0) aurora observed by the Swedish satellite MATS, EGUsphere [preprint], https://doi.org/10.5194/egusphere-2025-2324, 2025.

**A few minor comments are made below:**

• Section 5.8.4: to show the in-flight degradation of the UV channels, it would be preferable to show the calibration coefficient as a function of time over the whole dataset.

This is a good idea and we tried it. There is an indication of a trend in the absolute calibration coefficients for the UV channels, but we only have two months of data from the UV channels so it is difficult to be sure. Moreover, the measurements are made using four different stars that each are visible for only an even more limited time (one to four weeks). The data from each star are too short to have consistent trends, and if we combine the stars we cannot be certain that the trend is not due to the uncertainty of values reported in the star catalog. We will return to this when analysing the prolonged data set from MATS measurements and it may even be possible that we can correct for this in a later version of the data set. However, for the current version 1 of the data set (which the manuscript describes) any error relating to this is implicitly included in the estimate of the absolute calibration uncertainty, since it will result in a wider distribution of estimated calibration coefficients.

It should also be said that there is no indication of trend in the IR channels or in the relative calibration coefficients in either UV or IR.

A discussion on the above has been added to the manuscript.

• - Section 5.8.4 and Table 1: What is the database of star spectra used in the calibration? Please give the reference.

The databases are referred to in line 556 (558 in the updated manuscript), in section 5.8.3. The references for stellar spectrophotometry are not directly relevant for Table 1.

• - Lines 660-663: I am not convinced that the variation in polarization along the orbit is less than the variability in Rayleigh scattering due to variation in atmospheric density. Please justify with figures.

When handling polarisation sensitivity in the upcoming Level-2 analysis, it is not really critical whether signal variations due to polarisation are larger or smaller than signal variations due to atmospheric density. Nonetheless, we included that statement in the manuscript in order to illustrate for the reader that polarisation sensitivity is not a dominant effect. Here are some numbers behind our statement:

Along its sun-synchronous dawn/dusk orbit, MATS encounters solar scattering angles between  $70^{\circ}$  and  $110^{\circ}$ . In combination with the phase function for molecular Rayleigh scattering, this defines the polarisation of the scattered light reaching the limb instrument:  $P(1,70^{\circ}) / P(1,90^{\circ}) \gg (1+\cos^2(70^{\circ})) / (1+\cos^2(90^{\circ})) = 1.12$ . Invoking the polarisation sensitivity of the instrument (e.g., 15% for channel IR2), we find a variation of the measured signal by a factor  $\sim 1.1$  along the orbit. This calculation applies to single scattering of sunlight. In case of multiple scattering, involving upwelling radiation from the lower atmosphere, the polarisation effect is generally less.

The above factor 1.1 can be compared to the variation of total molecular Rayleigh, e.g. based on the variation of total atmospheric density as provided by MSIS. As an example, for June conditions and a tangent altitude of 90 km, this leads to a variation of the measured signal by a factor  $\sim$ 1.8 along the orbit.

**RC2 (Anonymous Referee #1, 28 Apr 2025)**

Citation: https://doi.org/10.5194/egusphere-2025-265-RC2

We thank the reviewer for his/her time and for useful comments, which we address below (original comment in italics).

This paper, for the most part, is very well written and is certainly relevant to AMT. I recommend the paper be published after a few minor issues, detailed below, are addressed.

One general comment is that I would consider using the term "uncertainty" and avoid the use of "error", especially when discussing random uncertainties. In statistics, "error" is often used as a specific term which essentially means a known bias that can be corrected for. If you're just using it in the generic sense of some deviation from the truth, "error" should be fine, but in this study it seems like what you're mostly discussing is uncertainties. There's a discussion of this in section 3.1 of the TUNER paper, <a href="https://doi.org/10.5194/amt-13-4393-202">https://doi.org/10.5194/amt-13-4393-202</a>.

We agree and have implemented this suggestion in the manuscript.

Line 7 – this sounds like 200-km in the direction of the orbit, is that what is meant? I would expect perpendicular to the orbital track. Maybe give both?

Yes, it is perpendicular to the track. The strips are continuous along-track. This has been clarified.

Lines 26-27 – "Firstly, gravity waves are relatively smallscale phenomena; thus, high-resolution observations are needed to detect them." Please give examples of what the authors deem to be "small" and "high".

This has been added.

Lines 35-38 – I recommend also discussing Aura/MLS when discussing other limb sounders as they also measure MLT temperatures and I believe are able to do tomographic retrievals.

We have added a short sentence pointing out that tomographic techniques have been used by other sensors to achieve horizontal resolutions on the order of 200 km

Lines 52-53 – this makes it sound like the 200 km swath is in the nadir direction, please rephrase

This has been rephrased.

Line 114 – when I first read this, I thought it was saying that smearing hasn't been accounted for in the current L0-L1 processing, but it seems like it is. Please rephrase

This has been rephrased.

Lines 180-182 – I'm not a python user, so I'm just assuming that "Scipy Rotation" is a Python package? I don't think this is the best way to describe how you do this step. Maybe

describe what is actually done by this "align\_vector" function. Honestly, the function you use might simply be an unnecessary detail.

We agree the text has been modified to remove the unnecessary detail.

Line 213 - I'm not seeing any mention of pixel solid angle in section 5.8.4, but rather in the intro of section 5.8.

This has been corrected.

Equations 6-7 – It's quite possible I'm missing something very obvious here, but shouldn't  $\Omega$  (units of sr) be in the denominator? If this equation is correct, wouldn't that give L units of photons sr/m2/s instead of what it should be, photons sr/m2/s? Please specify what the units are for all variables in these equations.

The reviewer is completely correct. Thank you! The definition of calibration coefficients was changed as the manuscript was written which led to mistakes in the formulas. They have now been corrected.

*Line 267 (and Fig 5 caption) – what is meant by "server"?*

It should be "severe". This has been corrected.

Figures 3-4 – no notes! I like them!

Thank you for the positive remark.

Figure 5 – Could be nice to see a panel with an example of an event-free capture beside this

Examples of image processing for "normal conditions" (i.e. outside the SAA) are given in Figures 3 and 4. Panels a) in those figures show typical images. While there are some SEE-free images, it is not uncommon to have one or two SEEs. The handling of those is described in the paper. The purpose of Figure 5 is rather to demonstrate the extent of the extreme SEE conditions in the SAA, giving an example of a heavy particle travelling at a shallow angle to the detector plane and leaving a significant charge deposited over multiple pixels.

*Line 313 – Should "darks" be "dark images"?*

Yes, that is correct. This mistake has been corrected.

Lines 346-348 – how often are natural signals expected to be strong enough that measurements will reach this saturation threshold?

We checked, and it turns out that that flag actually never was triggered in the whole data set. The non-linearity flag was triggered in approximately 0.3 % of the images for the channel with the strongest signal (IR2). The manuscript was updated to reflect this.

Figure 8 – I find everything about this figure confusing. The caption states, "the bottom of the MATS image is the top row of the matrix…" I don't understand why this choice for the figure was made, it seems arbitrarily confusing. Then "so that readout occurs downwards in the figure but upwards in the MATS image." But it appears that the readout occurs upwards in the figure and the main text clearly states that readout is downwards in the MATS image.

This next issue might just be a pdf interpreter issue, but I'm seeing the colours as yellow and blue, not red and green. Also, in the figure, the text "atmospheric signal not read out" is also confusing as it could mean either "atmospheric signal that is never read out" or "atmospheric signal yet to be read out". I would maybe call it something along this lines of "illuminated region" or "contaminated zone" or "low altitude, not measured". Although, even if all of this wasn't confusing, I don't think I would find this figure helpful and I'd probably recommend simply omitting it.

We agree that this figure was more confusing than educating, and have thus removed it.

*Line 617 – please define what is meant by "small"*

Estimates are given in the sections below. This has been made clearer.

Line 713 – "manifesting" should be "manifest"

This has been corrected.